# Ultrasound-Assisted Extraction of Antioxidants from *Baccharis dracunculifolia* and Green Propolis

**Renata Iara Cavalaro** **, Luis Felipe de Freitas Fabricio**
**and Thais Maria Ferreira de Souza Vieira \***

Department of Agri-Food Industry, Food & Nutrition, College of Agriculture Luiz de Queiroz,
University of São Paulo, Piracicaba 13418-900, São Paulo, Brazil; renata.cavalaro@usp.br (R.I.C.);
luisfreitas@usp.br (L.F.d.F.F.)
* Correspondence: tvieira@usp.br; Tel.: +55-19-3429-4150

**Abstract:** *Baccharis dracunculifolia* or rosemary-of-field is the principal botanical source used by Africanized bees *Apis mellifera* L. to produce green propolis in Southeastern Brazil. The phenolic compounds present in the plant and green propolis have been reported to be responsible for biological activities such as antioxidant capacity. This study aimed to optimize the ultrasound-assisted extraction of antioxidants compounds from rosemary-of-field using a central composite rotatable design (CCRD), and compare results to green propolis extract. An experimental design was performed to obtain responses of total phenolic content and antioxidant capacity. The results allowed observing that the optimum condition for both *Baccharis dracunculifolia* floral bud and raw green propolis antioxidant extraction was obtained with 99% ethanol solution. In this condition, Total Phenolic Content (TPC), Ferric Reducing Antioxidant Power (FRAP), and 2,2-diphenyl-1-picryl-hydrazyl (DPPH) values were 612.14 mg GAE. $g^{-1}$ sample, 534.39 µM ferrous sulfate $g^{-1}$ sample, and 72.37 µmol TEAC. $g^{-1}$ sample, respectively, for floral buds. These results have confirmed that optimization is a key step for effective and sustainable extraction processes to be feasible on an industrial scale. The proposed process can be easily adopted on a larger scale, as it uses very few inputs and presents straightforward steps, with the advantage of high efficiency in the extraction of phenolic compounds from the studied matrices compared to the results from the literature. The high concentration of antioxidants indicates that the products obtained can be considered as a sustainable bioactive source for food and cosmetic purposes.

**Keywords:** *Baccharis dracunculifolia*; green propolis; response surface methodology (RSM); optimization; ultrasound; antioxidant

## 1. Introduction

The interest of the food industries in replacing synthetic additives, focusing on products that contain bioactive compounds from natural sources and that are stable and functional, is rising [1–3]. Although efficient, some synthetic additives can present a toxic effect when used in high concentrations and also act as a pro-oxidant in contact with food [4]. These results reinforce the food system's need for effective natural antioxidants as an alternative to prevent food deterioration during processing and storage. Among the different types of propolis, green propolis is widely recognized in Brazil due to its diversity of biological activities such as antioxidant capacity. Green Propolis produced by Africanized *Apis mellifera* bees has a well-established botanical source known as *Baccharis dracunculifolia*, both of them have already been reported as potential substitutes for synthetic antioxidants intended for use as food additives and functional ingredients due to the potential of its bioactive compounds [5–7].

To produce green propolis, the *Apis Mellifera* bees act collecting resinous material from different plants, aiming to use this propolis to protect the hive from possible climate interference and external

invaders [8]. Green propolis is commonly found in the Southeast of Brazil, mainly between the states of São Paulo and Minas Gerais and its botanical source *Baccharis dracunculifolia* is considered an invasive plant due to its easy propagation [7,9]. The plant is commonly known by the regional names rosemary-of-field or vassourinha.

Both propolis and *Baccharis dracunculifolia* are known to present a diversity of compounds such as flavonoids and phenolic acids, and their antioxidant action is the result of the synergistic effect of these compounds [10]. However, while there are studies that approach the isolated optimization of green propolis or rosemary-of-field, there are no studies that evaluate and compare the optimization of the extraction process of bioactive compounds from the two sources, focusing on the use of ultrasound-assisted extraction. The optimization of processes using an experimental design allows exploring several factors that can influence the extraction process at the same time, using fewer samples and reducing the experimental error. An optimized extraction method renders it possible to accomplish efficiency in obtaining compounds of interest, which can directly affect environmental and economic impacts through the use of less toxic solvents, in lesser volumes and with reduced process time [11,12]. Associated with this statistical methodology, ultrasound can contribute to further reducing the time process and solvent consumption, and increasing the extraction yield [13]. The Central Composite Rotatable Design (CCRD) was successfully employed in determining the ideal conditions for the extraction of antioxidant compounds from *Z. lotus* fruits, under the influence of the ultrasound technique [14]. Data show that the use of ultrasound-assisted technology allows increasing the extraction yield by about 24%, specifically contributing to the increase in yield extraction of compounds with high antioxidant capacity, while decreasing the energy consumption used in the process [15]. Besides, it is considered a green technique, proposing to reduce solvent use, and, consequently, environmental pollution [16].

This work aimed to optimize the extraction of bioactive compounds from rosemary-of-field floral buds using solvent from a sustainable source in minimal processing time by ultrasound-assisted extraction. The activity of the extracts was compared to the performance of green propolis extract, attempting to determine the technical feasibility of exploiting the main vegetable raw material linked to the production of propolis.

## 2. Materials and Methods

### 2.1. Chemicals and Reagents

Folin–Ciocalteau reagent, ABTS (2.2′-azinobis-(3-ethylbenzothiazoline-6-acid)), Trolox (6-hydroxy-2,5,7,8-tetramethylchroman-2-carboxylic acid) and DPPH (2,2-diphenyl-1-picrylhydrazyl) were obtained from Sigma–Aldrich Chemical Co. (St. Louis, MO, USA). Other reagents were p.a. grade.

### 2.2. Baccharis dracunculifolia and Green Propolis

Samples of *Baccharis dracunculifolia* floral buds (rosemary-of-field) and raw green propolis were provided by the Brazilian company Natucentro Indústria e Apiários Centro Oeste Ltda. (19°59′46″ S, 45°48′38″ W) (Bambuí, Minas Gerais, Brazil). Both materials were collected in the same area during the flowering season (March/2018). Visual cleaning of the samples was carried out to remove any dirt. Then, the samples were vacuum packed and stored until analysis in a commercial freezer at −18 °C.

### 2.3. Experimental Design

A Central Composite Rotatable Design (CCRD) was carried out, aiming to study the effects of extraction time ($x_1$) and ethanol concentration ($x_2$) on the extraction of total phenolic content (TPC) and antioxidant capacity of rosemary-of-field extracts. Table 1 shows the ranges of the conditions for preparing the extracts and their levels of variation.

**Table 1.** Variation ranges for preparing rosemary-of-field extracts using an experimental design.

| Independent Variables | Variation Levels | | | | |
|---|---|---|---|---|---|
| | −1.41 | −1 | 0 | 1 | +1.41 |
| Time (minutes) | 80 | 92 | 120 | 148 | 160 |
| Ethanol (%) | 0 | 14.4 | 49.5 | 84.6 | 99 |

The general form of the regression equation used in this study is presented in the following Equation (1):

$$Y = \beta_0 + \beta_1 x_1 + \beta_2 x_2 + \beta_{12} x_1 x_2 + \beta_{11} x_1^2 + \beta_{22} x_2^2 + \varepsilon \tag{1}$$

where

$Y$ = response (dependent variable)

$x_1$ and $x_2$ = coded levels of explanatory variables

$\beta 0$ = mean

$\beta's$ = regression coefficients

$\varepsilon$ = residue

Rosemary-of-field extracts were produced according to the proposed design, with the study ranges presented in Table 1 at a fixed temperature of 35 °C, and analyzed for total phenolic content (TPC) and antioxidant capacity using the FRAP and DPPH methods. Response surfaces were generated to verify the trend of maximizing the yield of extracted bioactive compounds and, through the desirability figures, the optimal extraction point was determined. Subsequently, the results were compared to green propolis optimization data. Models were adjusted after linear, quadratic, and interactions effects evaluation. The adequacy of the models was determined by evaluating the *p*-value, besides to the coefficient of determination ($R^2$) and the Fisher test value (*F* value) obtained from the analysis of variance (ANOVA). Data were analyzed by the software Statistica 12.0 (Statsoft, USA). The statistical significance of the model parameters was determined at a 95% confidence level.

### 2.4. Baccharis dracunculifolia and Green Propolis Extract Production

The extraction processes occurred in an ultrasound bath with heating (LS Logen Scientific, LSUC2-120-3.0, Diadema, Brazil) at a frequency of 40 kHz. The rosemary-of-field floral buds extracts were obtained according to the experimental design and the resulting optimal condition. For green propolis, the extraction conditions used were previously optimized conditions by Cavalaro et al. [11]: 99% ethanol, sample to solvent ratio of 1:35 (mass: volume), during 20 min at 25 ± 3 °C. Extracts were centrifuged at 4695 g for 15 min in a refrigerated microcentrifuge (Hitachi Koki, Hitachinaka, Japan) and then they were filtered on Whatman n. 2. The supernatant was evaporated (Buchi Labortechnik AG rotor evaporator—model R-215, Buchi, Flawil, Switzerland) at 45 °C. Finally, the resulting solutions were stored refrigerated at 7 °C in amber flasks until the time of analysis.

### 2.5. Total Phenolic Content (TPC) Determination

The total phenolic content (TPC) was carried out according to Singleton, Orthofer, and Lamuela-Raventos (1999), establishing the gallic acid as the standard [17]. For the determination of the TPC, 0.5 mL of the extracts were used. Then, a volume of 2.5 mL of Folin-Ciocalteau reagent (diluted 1:10 *v/v*) and 2.0 mL of 4% $Na_2CO_3$ (*m/v*) were mixed. After 2 h of incubation in the dark at room temperature, absorbance was measured on a spectrophotometer (Shimadzu, model UV 1240, Japan) at 740 nm. A blank sample was prepared in the same way but without the sample. The results of TPC were expressed as gallic acid equivalents (mg GAE. $g^{-1}$ sample), calculated using a curve constructed with concentrations ranging from 2.5 to 40 µg. $mL^{-1}$.

### 2.6. Antioxidant Capacity Determination

#### 2.6.1. DPPH• (2,2-diphenyl-1-picryl-hydrazyl)

The antioxidant capacity by the DPPH method (2,2-diphenyl-1-picryl-hydrazyl) was determined according to Brand-Williams, Cuvelier and Berset (1995) [18]. For the analysis, 500 μL of different extracts dilutions were pipetted with 3.0 mL of ethanol and 300 μL of a 60 mM DPPH ethanolic solution [16]. The reduction in the DPPH radical was measured by monitoring the decrease in absorption at 515 nm using a mini-1240 UV spectrophotometer (Shimadzu, Kyoto, Japan) until stable extinction values (45 min) were obtained. The exact concentration of DPPH was calculated using a calibration curve. The radical scavenging activity of the extracts was calculated as the amount of DPPH inhibited by the sample when compared to a control. A standard curve was constructed with a Trolox solution (Trolox equivalent antioxidant capacity—TEAC) using concentrations from 20 to 200 μM and the DPPH radical elimination capacity was expressed in μmol TEAC. $g^{-1}$ sample. All determinations were performed in triplicate for each sample and the mean values and their corresponding standard deviations were provided.

#### 2.6.2. FRAP (Ferric Reducing Antioxidant Power)

Antioxidant activity by the FRAP method was performed according to what was addressed by Thaipong et al. (2006) with modifications [19]. Mixtures were prepared with 90 μL of the sample or standard solution, 270 μL of distilled water and 2.7 mL of the FRAP reagent, which was previously prepared with 0.3 M sodium acetate buffer, 20 mM ferric chloride solution and a solution of 10 mM TPTZ previously prepared with 40 mM HCl (10: 1: 1, *v/v/v*). In the case of the blank, only the FRAP reagent was used. The mixtures and the blank were prepared in a test tube and kept in a water bath with agitation at 37 ± 2 °C for 30 min. Then, the analysis was taken at 595 nm on the spectrophotometer (Shimadzu, model UV 1240, Japan). The standard curve was constructed using a standard solution of ferrous sulfate 2000 μM at concentrations of 300 to 1500 μM and the results were expressed in μmol ferrous sulfate. $g^{-1}$ sample.

### 3. Results

#### 3.1. Extraction Optimization

The extraction process of bioactive compounds from Baccharis dracunculifolia samples was directly affected by time ($x_1$) and ethanol concentration in solution ($x_2$) during ultrasound-assisted extraction. Besides, ultrasound proved to be efficient in releasing compounds trapped inside the cell to the hydroalcoholic solution, since the cavitation bubbles caused by the ultrasound mechanism induce the release of the active material [15]. Table 2 presents the results of total phenolic content (TPC) and antioxidant capacity (FRAP and DPPH), represented by the mean, followed by the standard deviation for each of the 12 runs of the experimental design.

As shown in Table 2, there was a tendency to raise the antioxidant power of extracts as the concentration of ethanol present in the solution increased. Therefore, it is important to note that in the first 120 min the variation on bioactive compounds occurred mainly due to the different concentrations of ethanol used in the extraction solution. TPC results ranged from 3.39 ± 0.02 to 612.14 ± 0.84 mg GAE. $g^{-1}$ sample and they were significantly affected by time (minutes) and ethanol concentration (%) ($p \leq 0.05$). The antioxidant capacity by the FRAP method varied from 24.23 to 534.39 μM ferrous sulfate. $g^{-1}$ sample, while by the DPPH method it varied from 0.24 to 72.37 μmol TEAC. $g^{-1}$ sample.

Table 3 presents the ANOVA and the coefficient of determination ($R^2$) obtained for TPC, FRAP, and DPPH responses. According to the F and *p* values, explanatory variables were selected to compose the models.

**Table 2.** Real values of the independent and dependent variables observed.

| | Independent Variables | | Dependent Variables | | |
|---|---|---|---|---|---|
| Essays | Time (min) | Ethanol (%) | TPC [1] | FRAP [2] | DPPH [3] |
| 1 | 92 | 14.4 | 13.47 ± 0.02 | 24.23 ± 0.02 | 0.24 ± 0.05 |
| 2 | 148 | 14.4 | 11.33 ± 0.13 | 26.73 ± 0.01 | 0.90 ± 2.47 |
| 3 | 92 | 84.6 | 116.1 ± 0.56 | 120.68 ± 0.04 | 14.61 ± 0.69 |
| 4 | 148 | 84.6 | 116.52 ± 7.15 | 238.18 ± 0.06 | 36.53 ± 0.49 |
| 5 | 80 | 49.5 | 30.57 ± 0.39 | 82.08 ± 0.01 | 14.84 ± 0.04 |
| 6 | 160 | 49.5 | 71.14 ± 0.73 | 121.15 ± 0.08 | 16.89 ± 0.92 |
| 7 | 120 | 0 | 3.39 ± 0.02 | 33.72 ± 0.10 | 0.54 ± 0.05 |
| 8 | 120 | 99 | 612.14 ± 0.84 | 534.39 ± 0.20 | 72.37 ± 0.80 |
| 9 | 120 | 49.5 | 91.97 ± 0.51 | 122.53 ± 0.12 | 12.65 ± 0.49 |
| 10 | 120 | 49.5 | 116.52 ± 0.28 | 113.46 ± 0.14 | 12.45 ± 0.39 |
| 11 | 120 | 49.5 | 120.52 ± 0.70 | 100.36 ± 0.08 | 13.22 ± 2.27 |
| 12 | 120 | 49.5 | 93.66 ± 0.74 | 112.41 ± 0.08 | 11.80 ± 1.18 |

[1] Result of (TPC) total phenolic content expressed in mg GAE. $g^{-1}$ sample; [2] Result of antioxidant capacity by FRAP (Ferric Reducing Antioxidant Power) method expressed in $\mu$M ferrous sulfate. $g^{-1}$ sample; [3] Result of antioxidant capacity by DPPH (2,2-diphenyl-1-picryl-hydrazyl) method expressed in $\mu$mol TEAC. $g^{-1}$ sample.

**Table 3.** Analysis of variance (ANOVA) for total phenolic content (TPC) and antioxidant capacity by the FRAP and DPPH method.

| Parameters | $R^2$ | Effects | Sum of Squares | *F* value | *p*-Value |
|---|---|---|---|---|---|
| TPC | 0.71 | | | | |
| Model | | - | 290036.0 | 162.28 | 0.001 * |
| $x_1$ | | 13.911 | 385.9 | 1.72 | 0.28 |
| $x_1^2$ | | −113.525 | 20449.5 | 91.53 | 0.002 * |
| $x_2$ | | 267.334 | 142510.1 | 637.89 | <0.0001 * |
| $x_2^2$ | | 144.919 | 33323.4 | 149.15 | 0.001 * |
| $x_1 x_2$ | | 1.282 | 1.6 | 0.007 | 0.93 |
| Error | | | 670.2 | | |
| Residue | | - | 94035.7 | - | - |
| Total | | - | 290706.2 | - | - |
| FRAP | 0.82 | | | | |
| Model | | - | 209679.6 | 316.52 | <0.0001 * |
| $x_1$ | | 43.9031 | 3843.5 | 46.41 | 0.006 * |
| $x_1^2$ | | −56.6150 | 5085.8 | 61.42 | 0.004 * |
| $x_2$ | | 254.2170 | 128868.0 | 1556.28 | <0.0001 * |
| $x_2^2$ | | 126.9228 | 25561.0 | 308.69 | <0.0001 * |
| $x_1 x_2$ | | 57.5012 | 3306.4 | 39.93 | 0.008 * |
| Error | | - | 248.4 | - | - |
| Residue | | - | 43263.3 | - | - |
| Total | | - | 209928.0 | - | - |
| DPPH | 0.83 | | | | |
| Model | | - | 4348.69 | 1566.39 | <0.0001 * |
| $x_1$ | | 6.38985 | 81.41 | 234.61 | <0.0001 * |
| $x_1^2$ | | −3.30945 | 17.37 | 50.07 | 0.005 * |
| $x_2$ | | 37.93784 | 2869.99 | 8270.15 | <0.0001 * |
| $x_2^2$ | | 17.40288 | 480.55 | 1384.75 | <0.0001 * |
| $x_1 x_2$ | | 10.63052 | 113.00 | 325.64 | <0.0001 * |
| Error | | - | 1.041 | - | - |
| Residue | | - | 787.41 | - | - |
| Total | | - | 4349.73 | - | - |

TPC = total phenolic content; FRAP = Ferric Reducing Antioxidant Power; DPPH = 2,2-difenil-1-picril-hidrazil.
* *p* value significant ($p \leq 0.05$).

ANOVA shows the impact of linear, quadratic, and interaction terms on the adjusted model. It was possible to observe that time and the interaction between time and percentage of ethanol were the only non-significant terms ($p \geq 0.05$), so they were excluded from the model represented by equation 2. However, the ethanol percentage had a significant and positive effect on the TPC response. The linear and quadratic terms for ethanol concentration during extraction accounted for the greatest effects, indicating that the increase in their values promoted an increase in the content of total phenolic compounds. The determination coefficient ($R^2$) obtained for TPC was considered satisfactory; being equal to 0.71 and the model's F value (162.28) implies that it is significant.

$$TPC = 106.01 - 56.76x_1{}^2 + 133.66x_2 + 72.45x_2{}^2 \qquad (2)$$

The responses of antioxidant capacity had the explained variation percentages due to the explanatory variables of 82% and 83% for FRAP and DPPH, respectively. Also, the F test was significant for the models ($p \leq 0.05$) (Equations (3) and (4)). Based on the analysis of the effects, all parameters (linear, quadratic, and interaction) showed a significant effect ($p \leq 0.05$) for both dependent variables, as shown in the mathematical models.

$$FRAP = 112.46 + 21.95x_1 - 28.30x_1{}^2 + 127.10x_2 + 63.46x_2{}^2 + 28.75x_1x_2 \qquad (3)$$

$$DPPH = 12.57 + 3.19x_1 - 1.65x_1{}^2 + 18.96x_2 + 8.70x_2{}^2 + 5.31x_1x_2 \qquad (4)$$

Through the response surface (Figure 1) the contribution of each variable to antioxidant capacity and TPC is evident. The three-dimensional response surfaces allow comprehending that an intermediate extraction time (120 min) by ultrasound was enough to produce extracts with higher phenolic compounds concentrations and antioxidant capacity. Besides this, extracts showed greater results on the maximum ethanol concentration tested (99%) for all responses (TPC, FRAP, and DPPH), indicating that the major compounds that were extracted had nonpolar characteristics.

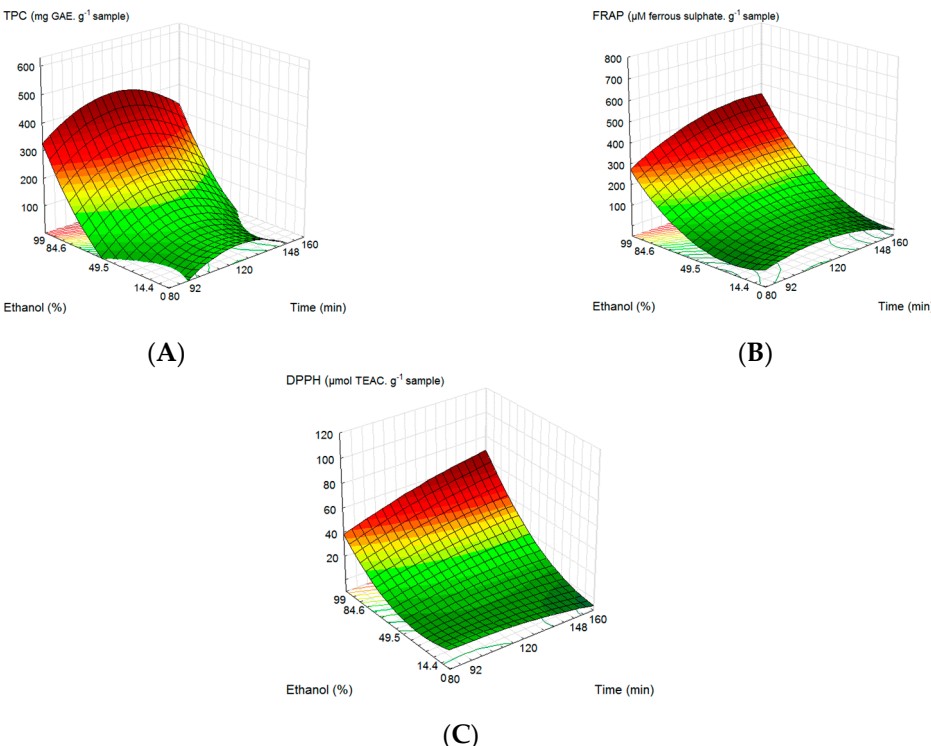

**Figure 1.** Response surfaces plots showing the effect of time and ethanol concentration on total phenolic content (TPC) and antioxidant capacity (FRAP and DPPH) represented by (**A**), (**B**) and (**C**), respectively.

The lowest result of TPC and antioxidant capacity were observed when 100% of the water was used for extraction, and the highest result was perceived at 99% ethanol solution as the solvent. Such results are in agreement with Casagrande et al. [5], who found in their study with Baccharis dracunculifolia that TPC was strongly influenced by the concentration of ethanol used in the extraction solution. Ethanol is considered ecologically correct and is preferably used for the extraction of antioxidant compounds since it harms less the environment when compared to other solvents such as methanol [20]. Martinez-Correa et al. [21] verified that the type of solvent used in the extraction influences the results of antioxidant capacity, due to each solvent extracting substances with different DPPH radical scavenging activities. Mokrani and Madani [22] also perceived this information at their work with pear, verifying that antioxidant compounds were strongly related to the type of solvent used.

Authors who studied the Baccharis dracunculifolia leaf as a source of phenolics found TPC values ranging from $69.00 \pm 2.00$ to $197.00 \pm 3.00$ mg GAE. $g^{-1}$ sample when they used several extraction processes, such as the application of supercritical carbon dioxide [21]. Such data exemplify the efficiency that ultrasound had in our study since it was possible to obtain higher results of TPC than those found by the authors mentioned above. Variation in TPC results had already been reported in the Baccharis dracunculifolia when different solvents in the extraction were used, obtaining results starting from 21.20 to 33.87 mg. $g^{-1}$ plant with the use of ethanol [5].

It is also necessary to check the antioxidant capacity by several methods, to identify the trend of extraction yield related to bioactive compounds. Therefore, FRAP and DPPH methods were used. In a study evaluating the antimicrobial and antioxidant capacity of B. oreophila essential oil, results of $145 \pm 0.02$ µmol TEAC 100 $mL^{-1}$ were found using the ABTS method, $71 \pm 0.03$ µmol TEAC 100 $mL^{-1}$ by DPPH and $409 \pm 0.04$ µM $FeSO_4$. $mL^{-1}$ according to the FRAP methodology [23]. The results are very close to those found in our work since the plants belong to the same Asteraceae family. Evaluating the total phenolic content and antioxidant capacity, it has observed that among the two native Uruguayan plants analyzed (Achyrocline satureioides and Baccharis trimera), Baccharis t. presented greater results and proved the effectiveness of the phenolic compounds found in this plant family [24].

The temperature used during our experiments with rosemary-of-field was fixed at 35 °C and it was suitable for extraction, avoiding the loss or degradation of active compounds. Hosseini et al. [25] studying rosemary's phenolic compounds extraction, found that 40 °C was sufficient to increase the mechanical effects of the ultrasound equipment without occurring compounds degradation.

Besides heating, the use of ultrasound-assisted extraction was essential to ensure antioxidant recovery, being a strategy considered efficient in the extraction of antioxidant compounds from leaves and buds of plants, such as Acacia confusa [26]. The results obtained in our study were also in agreement with Boudries et al. [27], who claimed to have obtained extracts from Capparis spinosa's buds rich in antioxidant compounds, mainly due to the efficiency of the ultrasound technique. The authors pointed out that this system allows high extraction yields in less processing time, being an ecological and "green" technique. Also, the extraction time was reported as one of the essential factors of its study using the ultrasound, since after a period there was no more increase in the content of phenolic compounds extracted. Data confirm what was addressed in our previous studies [11], which show that ultrasound requires less time to obtain high levels of total phenolic content compared to other extraction methods such as heat-assisted extraction and maceration, besides requiring less solvent [28,29]. Just like ultrasound, some techniques allow the process of extraction become greener while recovering the interested compounds, since they require less use of organic solvents, as approached by Doctor et al. with supercritical water extraction technology [30].

### 3.2. Baccharis dracunculifolia and Green Propolis Extract Production under Optimal Conditions

Optimum condition *Baccharis dracunculifolia* extraction was determined according to results of the total phenolic content, based on the figure of profiles for predicted values and desirability (Figure 2). This response was used to determinate the optimum point in a medium extraction time, defined in

our study to allow the enable an industrial application process, using mass production in the shortest possible extraction time with high productivity results.

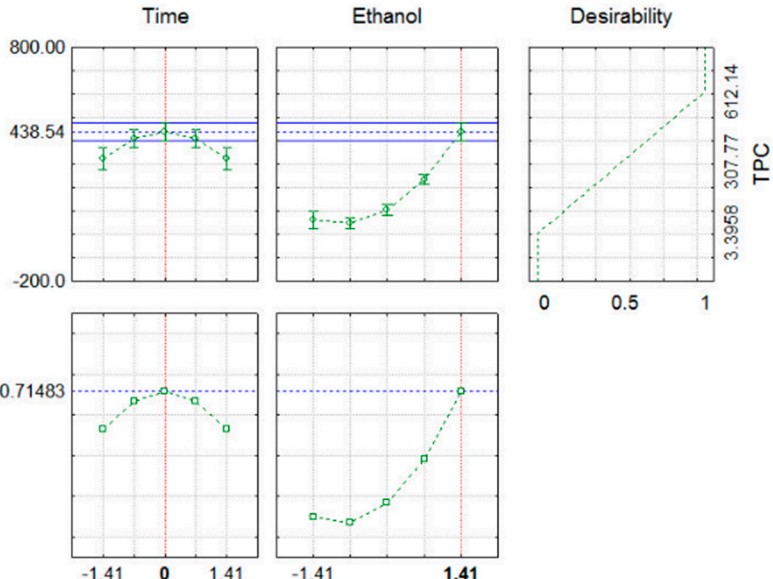

**Figure 2.** Profiles of predicted values and desirability obtained for the analysis of TPC.

The desirability function verified in Figure 2 is based on converting the response into a value ranging from 0 to 1. The value closest to 1 is attributed to the desired optimal response [31]. In the case of our study, the desirability remained at 0.71 and the optimum point made it possible to obtain significant results of antioxidant capacity when compared to data from the literature. It is noted through the profiles of predicted values and desirability that the optimum point of extraction of phenolic compounds was at 120 min, using 99% ethanol as the solvent concentration in solution. These conditions were tested and the results of TPC, FRAP, and DPPH were highly satisfactory. Total phenolic content was $612.14 \pm 0.84$ mg GAE. $g^{-1}$ sample and antioxidant capacity were $534.39 \pm 0.20$ μM ferrous sulfate. $g^{-1}$ sample by the FRAP method and $72.37 \pm 0.80$ μmol TEAC. $g^{-1}$ sample by DPPH. Thus, the model could be used for optimizing the active compounds from rosemary-of-field by ultrasound-assisted extraction coupled to an experimental design.

### 3.3. Comparison between Rosemary-of-Field and Green Propolis

In our study, the conditions to recover antioxidant compounds from green propolis resulted in extracts with TPC of 3462.4 GAE. $g^{-1}$ sample and antioxidant capacity of 18,652.9 μmol TEAC. $g^{-1}$ sample by the DPPH method and 36231.0 μM Ferrous Sulphate. $g^{-1}$ sample. The conditions were the same as used in our previous study according to Cavalaro et al. [11], which obtained as results: 1614.8 mg GAE. $g^{-1}$ sample of TPC and antioxidant capacity of 13244.5 μmol TEAC. $g^{-1}$ sample by the ORAC method and 13,412.1 μmol TEAC. $g^{-1}$ sample by ABTS method.

In this study, the results were superior to those previously reported and this can be explained by the seasonality of the green propolis since the samples were collected in rosemary-of-field flowering season (March) and in the previous paper, the samples were collected in November.

It was verified that despite the visible improvement in the extraction process effectiveness under optimized conditions, the products obtained from floral buds exhibited inferior antioxidant capacity to the green propolis extract. However, new plant-based studies should not be disregarded, since there is a growing demand towards vegetable-based diets and propolis is considered to be of animal origin [32]. Data were in agreement with what was found by Park et al. [33], which found that the antioxidant capacity of the green propolis ethanolic extract was superior to Baccharis dracunculifolia extracts using the DPPH method. Such authors reported that this difference in results probably occurred

due to the phenolic content present in the extracts. The total phenolic content is an indicator of the reducing power of a sample because it reflects the influence of several antioxidant compounds present, phenolic or not, in the test result [34]. It is plausible to comment that the period of higher frequency of resin collection by Africanized bees Apis mellifera L. coincides with the harvest of green propolis, therefore the phenolic compounds found in both materials are commonly similar. Bees prefer the Baccharis dracunculifolia plant due to its compatible composition, besides this plant has materials that are easily chewable by these insects [35].

## 4. Conclusions

Optimization using a CCRD to define the conditions of ultrasound-assisted extraction has enhanced the recovery of antioxidant compounds for both rosemary-of-field and green propolis. The extracts produced in our study presented higher antioxidant activity when compared to the results reported in the literature. Although the plant extract produced with ultrasound assistance present lower results of antioxidant capacity compared to green propolis, its vegetal origin allows studies to be further developed based on plant-based foods. Thus, both extracts developed in this study were suitable to produce natural ingredients with antioxidant capacity aiming food and cosmetic use.

**Author Contributions:** R.I.C.: Design of study, execution of the experiment, statistical analysis and writing, L.F.d.F.F.: Execution of experiment, T.M.f.d.S.V.: Acquisition of the financial support, responsible for research activity planning and execution, writing. All authors have read and agreed to the published version of the manuscript.

**Funding:** This work was supported by São Paulo Research Foundation—FAPESP (grant 2014/18227-4, and grant 2015/50437-1) and it was financed in part by the Coordenação de Aperfeiçoamento de Pessoal de Nível Superior—Brazil—CAPES (Finance Code 001).

**Acknowledgments:** Authors thanks the Natucentro Indústria e Apiários Centro Oeste Ltda for samples.

**Conflicts of Interest:** The authors declare no conflict of interest.

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
