# Peer review of "Ultrasound-Assisted Extraction of Antioxidants from Baccharis dracunculifolia and Green Propolis"

_processes, doi:10.3390/pr8121530_

Round 1

Reviewer 1 Report

  1. «The interest of the food industries in replacing synthetic additives, focusing on products that contain bioactive compounds from natural sources and that are stable and functional, is rising [1].»

It is necessary to submit a number of recently highly cited articles.

  1. «Propolis and its main botanical source, Baccharis dracunculifolia»

It should be indicated that this is for a specific region and a specific species of bees.

  1. It is not entirely clear from the text what is proposed for biotechnological use in the future, buds or propolis. Which of the presented types of raw materials is more technologically applicable in the future?
  2. The homogeneity of the collection site, environmental conditions can affect (albeit minor) on the composition of biologically active substances, and to a large extent on the overall concentration level. Thus, I would recommend specifying the collection sites and growing conditions. Probably, for a better correlation of the results, it would be better if the propolis was made from rosemary growing in the same area where the buds were collected for analysis.
  3. What caused the choice of the range of sample processing time?
  4. There is a great lack of an additional variable that would characterize the frequency characteristics of ultrasound. In any case (even if it is a fixed characteristic), the frequency should be specified.
  5. Why the extraction of propolis was not carried out according to the same plan as the extraction of razmarine buds, this would allow a better correlation between the results. Despite the fact that work on optimization of propolis extraction was carried out earlier (Cavalaro, RI; Cruz, RG; Dupont, S .; Moura Bell, JMLN; Vieira, TMFS In vitro and in vivo antioxidant properties of bioactive compounds from green propolis obtained by ultrasound-assisted extraction. Food Chem .: X 2019, 4, 100054.), the ranges of the extraction parameters there are different, in particular the time.
  6. «In addition, ultrasound proved to be efficient in releasing compounds trapped inside the cell to the hydro alcoholic solution.»

It is not entirely clear on the basis of what such a conclusion was made; it requires additional verification.

  1. Visually, the response surfaces for DPPH and FRAP do not reach their apical values at the maximum exposure time. This raises certain questions about the original design of the experiment.
  2. «The temperature used during our experiments with rosemary-of-field was fixed at 35 ° C and it was suitable for extraction.»

It would be logical to indicate this in the section on materials and methods.

  1. "Found that 40 ° C was sufficient to increase the mechanical effects of the ultrasound equipment."

I dare to suggest that a step of 5 degrees is critical in optimizing the extraction of biologically active substances.

  1. «The phenolic compounds identified by high performance liquid chromatography (results not shown in this article)»

I think it is not necessary to refer to the data not provided.

  1. The conclusion does not seem like an attempt to rethink the results

Author Response

Point 1: «The interest of the food industries in replacing synthetic additives, focusing on products that contain bioactive compounds from natural sources and that are stable and functional, is rising [1].» It is necessary to submit a number of recently highly cited articles.

Response 1: We appreciate your comments. We have added other references, which address the context about the growing interest of the food industry in natural additives between lines 31 - 33.

Point 2: «Propolis and its main botanical source, Baccharis dracunculifolia» It should be indicated that this is for a specific region and a specific species of bees.

Response 2: We added information about the type of bee, the botanical source, the type of propolis and the collection region throughout the Introduction. This is the Brazilian green propolis, found predominantly in the Southeast region, and produced by Africanized bees Apis mellifera, which uses exudates from the plant Baccharis dracunculifolia, as shown by Rodrigues, D. M., De Souza, M. C., Arruda, C., Pereira, R. A. S., & Bastos, J. K. (2020). The role of Baccharis dracunculifolia and its Chemical Profile on Green Propolis Production by Apis mellifera. Journal of Chemical Ecology, 46(2).

Point 3: It is not entirely clear from the text what is proposed for biotechnological use in the future, buds or propolis. Which of the presented types of raw materials is more technologically applicable in the future?

Response 3: In our study, we focused on the two raw materials, which present potential for the production of natural antioxidants from technologically feasible and environmentally friendly processes. In lines 325-329 we have addressed that the antioxidant capacity of green propolis is superior to that found for the plant Baccharis dracunculifolia.  However, it should be proposed that new studies regarding biotechnological activities using the plant, since it is an alternative of natural antioxidant specifically of vegetable origin and some food industries have directed their efforts to make products entirely with ingredients of vegetable origin.

Point 4: The homogeneity of the collection site, environmental conditions can affect (albeit minor) on the composition of biologically active substances, and to a large extent on the overall concentration level. Thus, I would recommend specifying the collection sites and growing conditions. Probably, for a better correlation of the results, it would be better if the propolis was made from rosemary growing in the same area where the buds were collected for analysis.

Response 4: The composition of phenolic compounds from the sources used in our study are variable according to seasonality. However, as approached by Figueiredo, S. M., S Binda, N., A Vieira-Filho, S., de Moura Almeida, B., RL Abreu, S., Paulino, N. & K Park, Y. (Physicochemical Characteristics of Brazilian Green Propolis Evaluated During a Six-Year Period. Current drug discovery technologies, 14(2), 2017), changes in composition are not considered relevant. As it was not the objective of our study to follow the influences of seasonal parameters, we cited only the collection site, the latitude, and longitude coordinates, as well as the month and year in which the samples were collected, as can be observed between lines 79 - 82. We also added to the text that both raw materials were collected in the same area, as suggested by the reviewer.

Point 5: What caused the choice of the range of sample processing time?

Response 5: We have applied 2 preliminary designs (pre-tests) varying the time and concentration of ethanol in the extraction solution until we reach the studied range to obtain the optimal extraction point. In the first pre-test, we varied the time from 5 to 45 minutes, and the ethanol concentration from 0 to 80%. In the second design, we increased the time, using the range of 30 - 90 minutes and keeping the ethanol concentration from 0 - 80%. By conducting the third design, presented in this manuscript, we were able to achieve the optimal condition according to the objectives of our study, varying the time from 80 - 160 minutes and the ethanol concentration from 0 - 99%.

Point 6: There is a great lack of an additional variable that would characterize the frequency characteristics of ultrasound. In any case (even if it is a fixed characteristic), the frequency should be specified.

Response 6: In our study, we worked with an Ultrasound bath with heating device (LS Logen Scientific, LSUC2-120-3.0, Brazil), without the possibility of frequency adjustments as can be done in some other types of equipment. This option was based on the need to control the extraction temperature for process optimization, since in equipment with probe there is an overheating, as observed in our preliminary tests and also explained by Chemat, F., Rombaut, N., Sicaire, A. G., Meullemiestre, A., Fabiano-Tixier, A. S., & Abert-Vian, M. (2017). Ultrasound-assisted extraction of food and natural products. Mechanisms, techniques, combinations, protocols, and applications. A review. Ultrasonics sonochemistry, 34.  We have included the frequency information in the manuscript.

Point 7: Why the extraction of propolis was not carried out according to the same plan as the extraction of razmarine buds, this would allow a better correlation between the results. Despite the fact that work on optimization of propolis extraction was carried out earlier (Cavalaro, RI; Cruz, RG; Dupont, S .; Moura Bell, JMLN; Vieira, TMFS In vitro and in vivo antioxidant properties of bioactive compounds from green propolis obtained by ultrasound-assisted extraction. Food Chem .: X 2019, 4, 100054.), the ranges of the extraction parameters there are different, in particular the time.

Response 7: Although Baccharis dracunculifolia is the main botanical source of green propolis, they are different materials and require different time, solvent concentration and extraction temperatures in order to obtain maximum recovery of its active compounds. If we used the same variables of the previous study (Cavalaro, RI; Cruz, RG; Dupont, S .; Moura Bell, JMLN; Vieira, TMFS In vitro and in vivo antioxidant properties of bioactive compounds from green propolis obtained by ultrasound-assisted extraction. Food Chem .: X 2019, 4, 100054.) we would not obtain an optimization of the extraction process for the plant, since the time used for green propolis was not sufficient to achieve satisfactory results.

Point 8: «In addition, ultrasound proved to be efficient in releasing compounds trapped inside the cell to the hydro alcoholic solution.» It is not entirely clear on the basis of what such a conclusion was made; it requires additional verification.

Response 8: The ultrasound efficiency is strongly related to the cavitation bubbles caused by the system, which are responsible to the erosion and therefore the exposure of the active material trapped in the cells. As the active material is exposed and the cell walls are broken down, the compounds of interest come into contact with the extracting solution and contribute to a significant increase in the concentration of phenolic compounds in the case of green propolis and Baccharis dracunculifolia. We have added an explanation on the subject in lines 162 - 164 of the article.

Point 9: Visually, the response surfaces for DPPH and FRAP do not reach their apical values at the maximum exposure time. This raises certain questions about the original design of the experiment.

Response 9: The overall objective of our study was to obtain satisfactory antioxidant capacity results in the shortest possible time. For this reason the selected condition was 120 minutes of extraction using 99% ethanol from the extraction solution, based on Total Phenolic Content (TPC) data.

Point 10: «The temperature used during our experiments with rosemary-of-field was fixed at 35 ° C and it was suitable for extraction.» It would be logical to indicate this in the section on materials and methods.

Response 10: We appreciate your comments. This information was added to the text on line 104.

Point 11: "Found that 40 ° C was sufficient to increase the mechanical effects of the ultrasound equipment." I dare to suggest that a step of 5 degrees is critical in optimizing the extraction of biologically active substances.

Response 11: We agree with your positioning. The paragraph was really confusing and that's why we rewrote it between lines 276 - 279.

Point 12: «The phenolic compounds identified by high performance liquid chromatography (results not shown in this article)» I think it is not necessary to refer to the data not provided.

Response 12: We thank you for your suggestion. This sentence was withdrawn from the article.

Point 13: The conclusion does not seem like an attempt to rethink the results

Response 13: We appreciate your comment, the conclusion was rewritten according to your previous suggestions.

Reviewer 2 Report

Overall its a well written manuscript. However there are some minor mistakes such as Line 75,109, Baccharis dracunculifolia should be italic. Also line 251, 'leaf' shouldnt be italic. Check like 266, 265, 263. 

another minor comment will be on they way you report your numbers. For example Line 307, '3.462,4' while at Line 311 '13412.1'. Perhaps good to follow one style throughout the article to ensure consistency.

TPC on Line 210 is written as 'TCP' 

For discussion, perhaps it will be great to discuss a little bit about the mechanism of Ultrasonic and why is it efficient in the extraction. 

Why is it DPPH and FRAP that you are using? Why are you not considering ABTS? What does the type of antioxidant that DPPH and FRAP detect? 

Author Response

Point 1: Overall its a well written manuscript. However there are some minor mistakes such as Line 75,109, Baccharis dracunculifolia should be italic. Also line 251, 'leaf' shouldnt be italic. Check like 266, 265, 263. 

 Response 1: We appreciate your comment, the corrections were accomplished.

Point 2: Another minor comment will be on they way you report your numbers. For example Line 307, '3.462,4' while at Line 311 '13412.1'. Perhaps good to follow one style throughout the article to ensure consistency.

Response 2: We appreciate your observation, the corrections were provided in the text.

Point 3: TPC on Line 210 is written as 'TCP' 

 Response 3: Actually the correct one is "TPC", we have corrected the error.

Point 4: For discussion, perhaps it will be great to discuss a little bit about the mechanism of Ultrasonic and why is it efficient in the extraction. 

Response 4: We strongly agree with your comment and so we have added a brief discussion on ultrasound extraction between lines 162 -164.

Point 5: Why is it DPPH and FRAP that you are using? Why are you not considering ABTS? What does the type of antioxidant that DPPH and FRAP detect? 

Response 5: The results of TPC, DPPH, and FRAP determinations reflect the antioxidant capacity of a sample by different principles on a comparable basis. They are methods with principles based on the oxy-reduction reaction between the oxidant and the antioxidant. In the case of DPPH, the method is based on the capacity of the DPPH radical to react with hydrogen donors, while the FRAP method is based on the capacity of phenols (represented by phenolic compounds in our study) to reduce Fe3+ into Fe2+. The ABTS was not used in this study because in previous studies we observed a high correlation between the results using DPPH, FRAP, ABTS, ORAC, and TPC, which based the selection of only 3 indicators in this study.

Reviewer 3 Report

I would like to thank you for giving me an opportunity to review your submission

# ISSN 2227-9717 to ‘MDPI’. This manuscript needs a minor revision. The scientific explanations and summary are satisfactory, however, there are few mistakes that were observed.

  1. Introduction

I would also suggest including the following reference in the introduction section with reference 13 on line 63 page # 2 of 13, to show the application of green extraction and application of ultrasonic and subcritical technique.

1). Doctor. N.; Kayan. B.; Parker, G.; Vang, K.; Yang. Y. Stability and extraction of vanillin and coumarin under subcritical water conditions. Molecules. 2020, 1-9.

  1. Materials and Methods

2.2 Baccharis dracunculifolia  and Green Propolis, line # 79-80 page # 2 of 13

(Then, the 80 samples were vacuum packed and stored until analysis in a commercial freezer at -18 ° C.)

Can you briefly explain how long it was stored into the freezer at -18 ° C?

  1. Materials and Methods

2.4 Baccharis and dracunculifola and green propolis extracts production, line 114, page # 3 of 13.

(Extracts were centrifuged at 4695 g for 15 minutes in 115 a refrigerated microcentrifuge (Hitachi KOKI; Japan)

Did you mean 4695 RPM?

  1. Results

3.1 Extraction optimization, line 157-158. Page# 4 of 13.

(The extraction process of bioactive compounds from Baccharis dracunculifolia samples was 158 directed)

Did you mean directly instead of directed?

  1. Results

3.1 Extraction optimization, line 206, page# 6 of 13.

(It 205 was possible to observe that time and the interaction between time and percentage of ethanol were 206 the only non-significant terms (p≥0.05)).

Space is required between p and ≥

  1. Results

3.1 Extraction optimization, line256-257, page 8 of 13.

(Variation in TPC results have already been reported in 256 the Baccharis dracunculifolia when it was used different solvents in the extraction, obtaining results 257 starting from 21.20 to 33.87 mg. g-1 plant with the use of ethanol [3].)

Have you observed any degradation with other solvents?

Author Response

 Point 1: I would also suggest including the following reference in the introduction section with reference 13 on line 63 page # 2 of 13, to show the application of green extraction and application of ultrasonic and subcritical technique.

 1). Doctor. N.; Kayan. B.; Parker, G.; Vang, K.; Yang. Y. Stability and extraction of vanillin and coumarin under subcritical water conditions. Molecules. 2020, 1-9.

Response 1: We thank you for your comment, we added a comment regarding green extraction as requested on lines 65 - 66. The article mentioned by the reviewer was also added in the discussion of the results on lines 291 - 294.

Point 2: 2.2 Baccharis dracunculifolia  and Green Propolis, line # 79-80 page # 2 of 13. (Then, the 80 samples were vacuum packed and stored until analysis in a commercial freezer at -18 ° C.). Can you briefly explain how long it was stored into the freezer at -18 ° C?

Response 2: In general, the articles do not address the specific time that the samples are stored in the freezer until the moment of analysis, as can be observed by Nour, V., Trandafir, I., & Cosmulescu, S. (2016). Optimization of ultrasound-assisted hydroalcoholic extraction of phenolic compounds from walnut leaves using response surface methodology. Pharmaceutical biology, 54(10). Our samples were stored around 2 months until the time of ultrasound-assisted extraction and analysis.

Point 3: 2.4 Baccharis and dracunculifola and green propolis extracts production, line 114, page # 3 of 13. (Extracts were centrifuged at 4695 g for 15 minutes in 115 a refrigerated microcentrifuge (Hitachi KOKI; Japan). Did you mean 4695 RPM?

Response 3: In fact, the programming of the equipment of rotations per minute (RPM) was not used, but of g-force, which is expressed in recent articles as "g", as we can see in the article by Rodriguez, E. S., Julio, L. M., Henning, C., Diehl, B. W., Tomás, M. C., & Ixtaina, V. Y. (2019). Effect of natural antioxidants on the physicochemical properties and stability of freeze-dried microencapsulated chia seed oil. Journal of the Science of Food and Agriculture, 99(4).

Point 4: 3.1 Extraction optimization, line 157-158. Page# 4 of 13. (The extraction process of bioactive compounds from Baccharis dracunculifolia samples was directed). Did you mean directly instead of directed?

Response 4: We appreciate the observation.  There was a typing error. We corrected it.

Point 5: 3.1 Extraction optimization, line 206, page# 6 of 13. (It was possible to observe that time and the interaction between time and percentage of ethanol were the only non-significant terms (p≥0.05)). Space is required between p and ≥

Response 5: Thanks for letting us know. The required spacing between p and ≥ has been corrected.

 Point 6: 3.1 Extraction optimization, line 256-257, page 8 of 13. (Variation in TPC results have already been reported in the Baccharis dracunculifolia when it was used different solvents in the extraction, obtaining results starting from 21.20 to 33.87 mg. g-1 plant with the use of ethanol [3].) Have you observed any degradation with other solvents?

Response 6: We did not evaluate any other solvent in our study, just ethanol for being considered an eco-friendly solvent. In general, studies indicate the degradation of phenolic compounds by the action of light, oxygen, and heating, as verified by Arruda, C., Ribeiro, V. P., Mejía, J. A. A., Almeida, M. O., Goulart, M. O., Candido, A. C. B., ... & Bastos, J. K. (2020). Green Propolis: Cytotoxic and Leishmanicidal Activities of Artepillin C, p-Coumaric Acid, and Their Degradation Products. Brazilian Journal of Pharmacognosy. Studies that address the degradation of Green Propolis and Baccharis dracunculifolia compounds through solvents were not found or presented in the literature.

Round 2

Reviewer 1 Report

The authors did a great job of finalizing the article. I hope that my comments helped them in this. Thank you very much for commenting on a number of points that were not clear to me, and possibly for future readers.
In my opinion, an article in this form may be allowed to be published.